# Survival features, prognostic factors, and determinants of diagnosis and treatment among Iranian patients with pancreatic cancer, a prospective study

**Mahdi Sheikh**[1,2], **Sahar Masoudi**[3], **Razieh Bakhshandeh**[3], **Alireza Moayyedkazemi**[3,4], **Farhad Zamani**[5], **Sepideh Nikfam**[3], **Masoumeh Mansouri**[3], **Neda Ghamarzad Shishavan**[3], **Saman Nikeghbalian**[6], **Paul Brennan**[2], **Reza Malekzadeh**[1,3], **Akram Pourshams**[1,3,7] *

1 Digestive Oncology Research Center, Digestive Diseases Research Institute, Tehran University of Medical Sciences, Tehran, Iran, 2 Section of Genetics, International Agency for Research on Cancer, World Health Organization, Lyon, France, 3 Liver and Pancreatobiliary Diseases Research Center, Digestive Diseases Research Institute, Tehran University of Medical Sciences, Tehran, Iran, 4 Department of Internal Medicine, Lorestan University of Medical Sciences, Khorramabad, Iran, 5 Gastrointestinal and Liver Diseases Research Center, Iran University of Medical Sciences, Tehran, Iran, 6 Department of Hepatobiliary Surgery, Shiraz University of Medical Sciences, Shiraz, Iran, 7 Gastrointestinal and Liver Diseases Research Center, Guilan University of Medical Sciences, Rasht, Iran

* akrampourshams@gmail.com

**Data Availability Statement:** All relevant data are within the paper and its Supporting Information files

## Abstract

### Objectives

Investigating the survival features, and determinants of treatment and stage at presentation in Iran

### Methods

461 patients with pancreatic ductal adenocarcinoma (PC) were prospectively enrolled from Shariati hospital, Tehran, Iran, between 2011–2018. All patients underwent endoscopic ultrasonography, computed tomography scanning, and physical examination. Validated questionnaire was completed for the participants and all were actively followed on monthly basis.

### Results

Median survival time was 6.5 months, and 1-, and 5-year survival rates were 26.2%, and 1.5%. Patients who were older (p<0.001), illiterate (p = 0.004), unmarried (p = 0.003), rural inhabitant (p = 0.013), opium user (p = 0.039), and had lower body mass index (BMI) (p = 0.002) had lower overall survival. Tumors located in the head of pancreas were more commonly diagnosed at lower stages (p<0.001). Only 10.4% of patients underwent surgery who were more commonly educated (p<0.001), married (p = 0.005), had a tumor located in the head of pancreas (p = 0.016), and were diagnosed at lower stages (p<0.001). After adjustment for potential confounders and risk factors, rural inhabitance (HR: 1.33 (95% CI: 1.01–1.74)), having more symptoms (HR for each increasing symptom: 1.06 (1.02–1.11)), using

**Funding:** This project was funded by the National Institute for Medical Research Development (grant number: 963236, grant recipient: Akram Pourshams). MS was supported by the World Cancer Research Fund International (grant number: WCRF 2016/1633, grant recipient: Paul Brennan). Funding for grant 2016/1633 was obtained from Wereld Kanker Onderzoek Fonds (WKOF), as part of the World Cancer Research Fund International grant programme.

**Competing interests:** The authors have declared that no competing interests exist.

opium (HR: 1.51 (1.04–2.20)), having a tumor located in the body of pancreas (HR: 1.33 (1.02–1.75)), and having an advanced tumor stage (HR: 2.07 (1.34–3.19)) remained significantly associated with increased risk of mortality. After the adjusting for potential confounders, we did not find significant relationships between smoking, alcohol intake, and BMI with the risk of death among patients with pancreatic cancer.

## Conclusions

Iranian patients with PC have very poor long-term survival. Besides tumor's stage and location, socioeconomic disparities could affect the probabilities of receiving treatment and/or survival in these patients. Opium use is an independent risk factor for mortality among PC patients in Iran.

## Introduction

In 2018, an estimated 458,918 people across the world were diagnosed with pancreatic cancer (PC), and 432,242 died of this disease, making it the 7th most common cause of cancer death worldwide [1]. Poor prognosis remains one of the main characteristics of PC [1]; The burden of PC is higher in the developed countries where the 5-year survival rates remain around 3–9% without significant changes over the past decades [2–8].

With increasing longevity, socioeconomic developments, and lifestyle changes in the developing countries a substantial increase in the incidence of PC is expected in these regions [2, 3]. Currently, there is a significant lack of knowledge on the survival features and prognostic factors of PC in the developing countries. Most available information originate from retrospective studies with limited sample size that are prone to different biases and are not generalizable [9, 10]. Further, we have previously identified using opium as a novel PC risk factor in populations where recreational use of opium is widespread [11, 12], however, it is not known if using opium can also affect the survival of PC patients. Opium consumption was recently categorized as a carcinogen to humans and opium pyrolysates were confirmed to have genotoxic effects by the International Agency for Research of Cancer (IARC) [13]. Therefore, investigating the effect of opium use on the survival of PC patients is particularly important due to the widespread use of opium and its derivatives (opiates) for managing cancer pain in the developing regions.

In this prospective study we analyze the clinical, pathological, therapeutic, and survival features of 461 histologically confirmed PC cases to provide reliable information on different features of PC in Iranian patients and to address the unmet medical needs and the possible shortfalls in the management of PC in this region. We further assessed whether there is any associations between opium consumption and risk of death among PC patients.

## Materials and methods

### Study population and design

This prospective study was performed on the patients with pancreatic adenocarcinoma that were originally recruited to a large case-control study aimed to investigate the epidemiologic features, clinical characteristics, and risk factors of pancreatic cancer in Iranian patients [11]. The detailed methods for this study have been previously described [11, 14]. Briefly, participants were recruited from Shariati hospital (a tertiary referral hospital) in Tehran, Iran,

between January 2011 and January 2018. Patients who were suspicious for having a pancreatic mass and were referred for performing endoscopic ultrasonography (EUS) with biopsy from pancreatic tissue were invited to participate in the original case-control study. Upon enrolment an informed written consent was obtained from the participants, then a validated and reliable questionnaire was used to collect detailed information on demographics, socioeconomic status, lifestyle and different exposures [14]. The participants were then referred for collection of bio-samples and performing EUS, and in case of finding a mass or cystic lesions, they underwent fine needle aspiration (FNA). The obtained samples were then reviewed by one expert pathologist who was blinded to the questionnaire data. If the diagnosis of ductal adenocarcinoma could not be finalized by hematoxylin/eosin staining, the samples were evaluated using an immunohistochemistry panel to differentiate ductal adenocarcinoma from other tumors. All participants who had histologically confirmed pancreatic ductal adenocarcinoma were included in this study and were actively followed. This study was approved by the Institutional Review Board and Ethics Committees of the Digestive Disease Research Institute of Tehran University of Medical Sciences, and the National Institute for Medical Research Development (approval number: IR NIMAD REC 1396 139).

## Questionnaire data upon enrolment

Upon enrolment all participants were visited and interviewed by a trained general physician who completed a detailed questionnaire for each participant, and performed a brief physical examination and anthropometric measurements. Body mass index (BMI) was calculated at enrolment for each participant by dividing the measured weight (kilograms) by the measured height (meters) squared. We categorized the participants based on their BMI at enrolment as: underweight (BMI < 18.5), normal ($18.5 \leq$ BMI < 25), overweight ($25 \leq$ BMI < 30), and obese (BMI $\geq$ 30).

The questionnaire contained 113 questions, collecting data on demographics, medical history, different exposures, and signs and symptoms of the current illness. The validity and reliability of this questionnaire was previously confirmed in the pilot phase of this study [14].

The questionnaire included questions about regular consumption of opium, cigarettes, and alcohol, the starting and ending ages for using each agent, and the frequency and amount of consuming each agent. Regular use of opium and cigarettes were defined as using these agents at least once per week for six consecutive months, while regular alcohol drinking was defined as drinking alcohol at least once per month for six consecutive months. For opium use the participants were further asked about the route (smoking/ingestion) and type (raw opium (teriak) / refined opium (shireh), opium dross (sukhteh)) of used opium. The detailed description of this opium types are presented in S1 File. Most opium users in this study (93%) reported using only raw opium, therefore we did not separate the analyses based on opium types. To assess the effects of using opium we categorized the participants as "never opium users", "former opium users", and "current opium users". To assess the effects of smoking cigarettes we categorized the participants as "never smokers", "former smokers", and "current smokers". Similarly, to assess the effects of drinking alcohol we categorized the participants as "never alcohol drinkers", "former alcohol drinkers", and "current alcohol drinkers". Former opium users/ smokers/alcohol drinkers included participants who had previously used the corresponding agent and had quitted using opium/smoking cigarettes/drinking alcohol at least for one year before the interview.

The patients were interviewed before undergoing EUS and knowing the final diagnosis to minimize the potential responder and interviewer bias that might possibly happen after identifying the case status of the participants, and also to avoid receiving possible inaccurate

responses due to EUS sedation. After performing the EUS, the endosonographist was asked to complete the questionnaire data regarding the presence of any tumor in the pancreas, the size and location of the tumor, and also any involvement of the vascular and lymphatic tissues.

All medical, imaging, and pathology documents were reviewed by an expert gastroenterologist to verify the stage of PC. Staging was performed using the information from EUS and conventional computed tomography (CT) scan that were available on enrolment (before any medical or surgical interventions) based on the TNM classification method that is proposed by the American Joint Committee on Cancer (AJCC) (S2 File) [15].

## Follow-up process

After confirming the diagnosis of pancreatic adenocarcinoma, the patients were introduced to a multidisciplinary team that included gastroenterologist, oncologist, radiotherapist, radiologist, pathologist, and surgeon to receive the optimal management and treatments. All patients have been actively followed through monthly telephone contacts to ascertain their vital status and collect updated information on any new medical and therapeutic interventions they had been receiving. In case of any new update, a team was assigned to collect copies of all newly performed medical procedures, laboratory tests, imaging studies, treatment details, surgery reports, and surgical pathology reports through contacting the patients, their relatives, and the corresponding medical centers. The gathered information was then monthly reviewed by an expert gastroenterologist to complete the follow-up data.

## Statistical analysis

We used the Kaplan-Meier method to calculate the survival probabilities and construct the survival curves, and used the log-rank test to assess the differences in Kaplan-Meier estimates. Cox proportional hazards regression models were used to estimate hazard ratios (HRs) and corresponding 95% confidence intervals (CIs) for assessing the prognostic effects of different exposures on the survival of PC patients. The entry time was defined as the date at which the participant was diagnosed with PC, and the exit time was the end of follow-up time, defined as the date of death for those patients who died during the follow-up, and the date of last follow-up for those who were still alive through the last follow-up on July 06, 2019.

The associations between different demographical, clinical, and tumoral characteristics with PC stage and the received treatment were analyzed using the Chi-square test. The effects of demographical, clinical, and tumor characteristics on the survival time were tested using two multivariate Cox regression models. The first multivariate model (model-1) was adjusted for the demographics including age (continuous), sex (male/female), formal education (ever/never), marital status (married/single), and residence (urban/rural), while the second multivariate model (model-2) further included BMI at enrolment (underweight/normal/overweight/obese), number of symptoms upon diagnosis (continuous), smoking cigarettes (never/former/current), using opium (never/former/current), drinking alcohol (never/former/current), tumor location (head/body/tail), and stage of PC (I/II/III/IV). The final models were further stratified by the received treatments (palliative / chemotherapy / surgery) that violated the proportional hazard (PH) assumptions by showing time-varying effects. The PH assumption was tested using Schoenfeld's global test.

To assess reverse causality of opium use, we repeated the analysis after excluding all participants who started using opium during the last two years before diagnosis. All statistical analyses were two-sided and performed using Stata statistical software version 14 (Stata Corporation, College Station, Texas, USA).

## Results

### Demographics

Four-hundred and sixty-one patients with histologically confirmed pancreatic ductal adenocarcinoma were enrolled in this prospective study. All patients were successfully followed. The median survival time in this study was 6.5 months with a range of <1 month to 89.1 months, while the 1-, 3-, and 5-year survival rates were 26.2%, 4.3% and 1.5%, respectively. The mean ± SD of the participants' age was 64.1 ± 11.5 years. Most participants were male (61.1%), had formal education (59.4%), were married (81.7%), were overweight/obese (55.2%), and lived in urban areas (80.0%). Of the participants, 36.5% reported ever smoking, 16.1% reported ever use of opium, and 10.2% reported ever consumption of alcohol (Table 1).

Higher overall survival was observed in patients who were younger (p<0.001), educated (p = 0.004), married (p = 0.003), overweight/obese (p = 0.002), lived in urban areas (p = 0.019), and had never used opium (p = 0.039) (Table 1 and Fig 1).

### Clinical symptoms

Details of clinical symptoms upon diagnosis are listed in S1 Table. Abdominal pain (82.4%) and unintentional weight loss (81.3%) were the most common presenting symptoms. Further, more than half of the patients had dark-colored urine (55.7%) and/or jaundice (52.2%). Of the whole patients, 6.7% had presented with only one symptom, while 50% had at least six symptoms upon diagnosis (S1 Table).

Lower overall survival was observed in patients who had abdominal pain (p<0.001), unintentional weight loss (p = 0.002), and constipation (p = 0.02) upon diagnosis. Having more symptoms was also associated with lower overall survival (p = 0.013) (S1 Table).

### Tumor and treatment characteristics

Most tumors (74.1%) originated from the head of pancreas. Of the participants, 9.5% were diagnosed at stage I, 54.1% at stage II, 17.7% at stage III, and 18.6% were diagnosed at stage IV of PC. Surgical resection was performed on 10.3% of the participants, while 56.7% received chemotherapy either with surgery (adjuvant chemotherapy) (11.2% of those who received chemotherapy) or without surgery (88.8% of those who received chemotherapy). 40.3% of the participants only received palliative treatment (Table 1).

Higher overall survival was observed in patients who were diagnosed at lower stages (p<0.001), those who received surgical resection of the tumor, and those who received chemotherapy (p<0.001) (Table 1).

### Determinants of stage and the probability of receiving treatment

We did not find significant associations between demographics, smoking, opium use, and alcohol intake with the stage of PC upon diagnosis (Table 2). Tumors that originated in the head of pancreas were more commonly detected at lower stages, while tumors located in the body and tail of pancreas were more commonly detected at higher stages (p<0.001) (Table 2). Having abdominal pain was more common in patients who were diagnosed higher stage tumors (p = 0.030). In contrary, having jaundice (p<0.001), dark-colored urine (p<0.001), and pruritus (p<0.001) were more common among patients who were diagnosed with lower stage tumors (Table 3).

Education and marital status were the only demographical factors that were associated with the type of received treatment. Patients who were educated (p<0.001) and married (p = 0.005) were more likely to receive surgery and/or chemotherapy than the illiterate and single patients

**Table 1. Estimates of overall survival rates by demographics and tumor characteristics among Iranian patients with pancreatic ductal adenocarcinoma.**

| Characteristics | No (%) | Median Survival (month) | P-value ¶ | 1 year Survival % | 3 years survival % | 5 years survival % |
|---|---|---|---|---|---|---|
| **Total** | 461 (100) | 6.5 | | 26.2 | 4.3 | 1.5 |
| **Age (years)** | | | <0.001 | | | |
| <60 | 153 (33.1) | 7.8 | | 33.9 | 7.8 | 3.2 |
| 60–70 | 160 (34.7) | 6.7 | | 25.6 | 4.3 | 0.6 |
| >70 | 149 (32.1) | 5.2 | | 18.9 | 0.6 | 0.0 |
| **Gender** | | | 0.36 | | | |
| Male | 282 (61.1) | 6.1 | | 25.1 | 3.9 | 0.7 |
| Female | 179 (38.8) | 6.9 | | 27.9 | 5.0 | 2.7 |
| **Formal education** | | | 0.004 | 21.9 | 3.2 | 2.6 |
| Never | 187 (40.5) | 5.2 | | 29.2 | 5.1 | 0.7 |
| Ever | 274 (59.4) | 7.2 | | | | |
| **Marital status** | | | 0.003 | | | |
| Married | 377 (81.7) | 6.8 | | 28.1 | 5.3 | 1.8 |
| Single | 84 (18.2) | 5.8 | | 17.8 | 0.0 | 0.0 |
| **Residence** | | | 0.013 | | | |
| Urban | 369 (80.0) | 6.8 | | 28.1 | 4.6 | 1.9 |
| Rural | 92 (19.9) | 5.2 | | 18.4 | 3.2 | 0.0 |
| **Body mass index** * | | | 0.002 | | | |
| Underweight | 12 (2.6) | 4.4 | | 13.0 | 0 | 0 |
| Normal | 190 (42.1) | 5.8 | | 20.7 | 6.0 | 2.4 |
| Overweight | 166 (36.8) | 8.3 | | 34.7 | 8.3 | 3.2 |
| Obese | 83 (18.4) | 6.9 | | 35.4 | 18.1 | 12.5 |
| **Smoking status** | | | 0.88 | | | |
| Never | 293 (63.5) | 6.6 | | 26.9 | 3.7 | 2.0 |
| Former | 59 (12.8) | 6.8 | | 27.1 | 3.3 | 0.0 |
| Current | 109 (23.6) | 6.1 | | 23.8 | 6.4 | 0.9 |
| **Opium use status** | | | 0.039 | | | |
| Never | 387 (83.9) | 6.9 | | 27.9 | 4.6 | 1.8 |
| Former | 15 (3.2) | 4.6 | | 20.0 | 0.0 | 0.0 |
| Current | 59 (12.8) | 4.9 | | 16.9 | 3.3 | 0.0 |
| **Alcohol use status** | | | 0.51 | | | |
| Never | 414 (89.80) | 6.5 | | 25.6 | 3.8 | 1.4 |
| Former | 24 (5.21) | 4.9 | | 25.0 | 12.5 | 4.1 |
| Current | 23 (4.99) | 8.4 | | 39.1 | 4.3 | 0.0 |
| **Tumor location** | | | 0.47 | | | |
| Head | 342 (74.1) | 6.8 | | 28.6 | 5.2 | 1.7 |
| Body | 102 (22.1) | 6.2 | | 19.6 | 0.9 | 0.0 |
| Tail | 17 (3.6) | 6.0 | | 17.6 | 5.8 | 5.8 |
| **Tumor stage at diagnosis** | | | <0.001 | | | |
| I | 44 (9.5) | 9.5 | | 38.6 | 9.0 | 6.8 |
| II | 249 (54.1) | 7.1 | | 27.7 | 5.6 | 1.2 |
| III | 82 (17.7) | 7.2 | | 29.2 | 1.2 | 0.0 |
| IV | 86 (18.6) | 3.4 | | 12.7 | 1.1 | 1.1 |
| **Primary treatment** | | | <0.001 | | | |
| Palliative | 186 (40.3) | 2.9 | | 8.0 | 0.0 | 0.0 |
| Chemotherapy | 227 (49.2) | 8.7 | | 33.0 | 3.9 | 1.3 |
| Surgical resection (+/- chemotherapy) | 48 (10.4) | 19.7 | | 64.5 | 22.9 | 8.3 |

¶ Log-rank test P value

* Data on body mass index at enrolment is missing for 10 participants

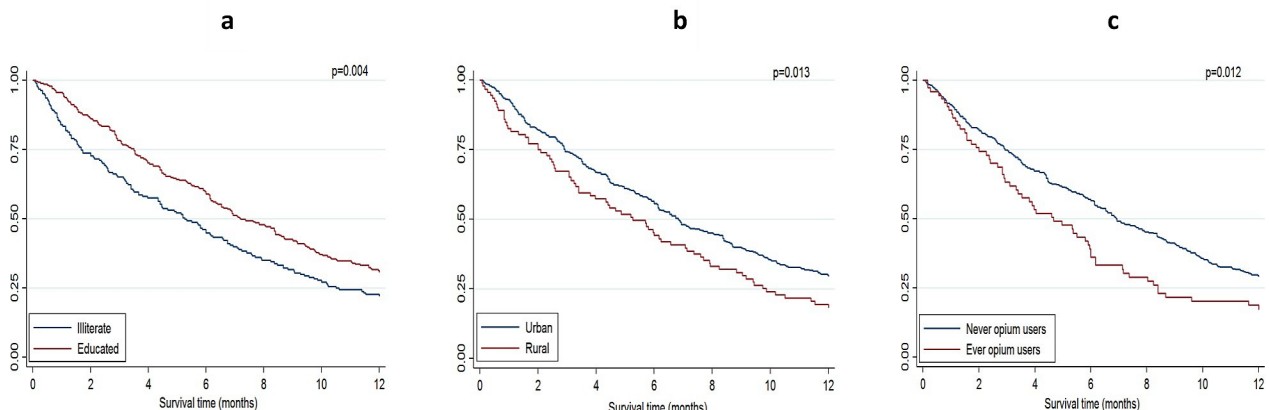

**Fig 1. Overall survival during the first 12 months of diagnosis in Iranian patients with pancreatic ductal adenocarcinoma.** The survival curves are stratified by a. having formal education, b. residence, c. ever use of opium.

(Table 4). Surgical resection was more commonly performed on the tumors that were diagnosed at early stages (p<0.001), and were located in the head of pancreas (p = 0.016) (Table 4).

## Prognostic factors of pancreatic cancer survival

After adjustment for all potential confounders and risk factors, rural inhabitance (HR: 1.33, 95%CI: 1.01–1.74), having more symptoms at diagnosis (HR for each increase in the number of symptoms: 1.06, 95%CI: 1.02–1.11), current opium use (HR: 1.51, 95%CI: 1.04–2.20), having a tumor that is located in the body of pancreas (HR: 1.33, 95%CI: 1.02–1.75), and having an advanced tumor stage (HR: 2.07, 95%CI: 1.34–3.19) remained significantly associated with increased risk of mortality in patients with pancreatic ductal adenocarcinoma (Table 5).

Only three participants had started using opium in the last two years before diagnosis and excluding these patients did not affect the observed results.

## Discussion

This prospective study of 461 patients with histologically confirmed pancreatic ductal adenocarcinoma showed a very poor long-term survival among Iranian patients with PC. The 5-year survival rates even in those who underwent surgical resection of the tumor were lower than other countries. Tumor stage at diagnosis showed some associations with tumor location and clinical symptoms. While, the type of received treatment differed across the strata of education and marital status. After adjustment for potential confounders, rural inhabitance, having more symptoms at diagnosis, current use of opium, having a tumor located in the body of pancreas, and having an advanced tumor stage remained significantly associated with increased mortality risk among PC patients.

The median survival time in this study was 6.2 months, and the 1- and 5-year survival rates were 26.2% and 1.5%, respectively. A survival analysis that included 202,584 European individuals who were diagnosed with pancreatic cancer between 2000 and 2007, showed a median survival time of 4.6 months, and an average 1- and 5-year survival rates of 25.9%, and 6.9% across Europe [16, 17]. One- and five-year survival for patients diagnosed with pancreatic cancer during 2010–2011 in England and Wales, were 20.8% and 3.3%, respectively [5], while 1- and 5-year survival rates for patients who were diagnosed with pancreatic cancer during 2009–2015 in the United States were 28% and 9.3%, respectively [18]. Finally, a recent analysis of 1,229,379 adults from 290 registries in 59 countries, showed that the 5-year net survival estimates for PC were generally in

**Table 2. Associations between different demographics and tumor characteristics with the tumor stage at diagnosis among Iranian patients with pancreatic ductal adenocarcinoma.**

| Characteristics | Stages I & II (total n = 293) N (%) | Stages III & IV (total n = 186) N (%) | P-value ¶ |
|---|---|---|---|
| **Gender** | | | 0.36 |
| Male | 174 (59.3) | 108 (64.2) | |
| Female | 119 (40.6) | 60 (35.7) | |
| **Formal education** | | | 0.67 |
| Ever | 172 (58.7) | 102 (60.7) | |
| Never | 121 (41.3) | 66 (39.2) | |
| **Marital status** | | | 0.51 |
| Married | 237 (80.8) | 140 (83.3) | |
| Single | 56 (19.1) | 28 (16.6) | |
| **Residence** | | | 0.72 |
| Urban | 236 (80.5) | 133 (79.1) | |
| Rural | 57 (19.4) | 35 (20.8) | |
| **Body Mass Index** * | | | 0.39 |
| Normal | 8 (2.7) | 4 (2.4) | |
| Underweight | 124 (43.2) | 66 (40.2) | |
| Overweight | 109 (37.9) | 57 (34.7) | |
| Obese | 46 (16.0) | 37 (22.5) | |
| **Smoking** | | | 0.87 |
| Never | 188 (64.1) | 105 (62.5) | |
| Former | 38 (12.9) | 21 (12.5) | |
| Current | 67 (22.8) | 42 (25.0) | |
| **Opium use** | | | 0.26 |
| Never | 247 (84.3) | 140 (83.3) | |
| Former | 12 (4.1) | 3 (1.7) | |
| Current | 34 (11.6) | 25 (14.8) | |
| **Alcohol drinking** | | | 0.65 |
| Never | 266 (90.7) | 148 (88.1) | |
| Former | 14 (4.7) | 10 (5.9) | |
| Current | 13 (4.4) | 10 (5.9) | |
| **Tumor location** | | | <**0.001** |
| Head | 249 (84.9) | 93 (55.3) | |
| Body | 39 (13.3) | 63 (37.5) | |
| Tail | 5 (1.7) | 12 (7.1) | |

¶ P value for chi-squared test

* Data on body mass index at enrolment is missing for 10 participants

the range 5–15% throughout 2000–2014 [19]. The median survival time and 1-year survival rates in our study are comparable to the reported rates from the developed countries. However, despite the high percentage of participants who were diagnosed at early stages, the 5-year survival rates in our study are lower than the reported rates in other countries. These results show the need for more efforts to improve the clinical management of PC patients in this region and to design more studies aiming to investigate the underlying reasons of the observed poor long-term survival in PC patients with early stage tumors and those who undergo surgical resection.

In the current study several indicators of socioeconomic status were related to survival and the probability of receiving treatment in PC patients. Patients who lived in rural areas, were

**Table 3. Associations between different clinical symptoms with the tumor stage at diagnosis among Iranian patients with pancreatic ductal adenocarcinoma.**

| Characteristics | Stages I & II (total n = 293) N (%) | Stages III & IV (total n = 186) N (%) | P-value ¶ |
|---|---|---|---|
| **Abdominal pain** | | | **0.030** |
| No | 60 (20.4) | 21 (12.5) | |
| Yes | 233 (79.5) | 147 (87.5) | |
| **Unintentional weight loss** | | | 0.24 |
| No | 60 (20.4) | 27 (16.07) | |
| Yes | 233 (79.5) | 141 (83.9) | |
| **Dark-colored urine** | | | **<0.001** |
| No | 111 (37.8) | 93 (55.3) | |
| Yes | 182 (62.1) | 75 (44.6) | |
| **Jaundice** | | | **<0.001** |
| No | 116 (39.5) | 104 (61.9) | |
| Yes | 177 (60.4) | 64 (38.1) | |
| **Light-colored stool** | | | 0.081 |
| No | 164 (55.9) | 108 (64.2) | |
| Yes | 129 (44.0) | 60 (35.7) | |
| **Constipation** | | | 0.57 |
| No | 177 (60.4) | 97 (57.7) | |
| Yes | 116 (39.5) | 71 (42.2) | |
| **Anorexia** | | | 0.14 |
| No | 170 (58.0) | 109 (64.8) | |
| Yes | 123 (41.9) | 59 (35.1) | |
| **Pruritus** | | | **<0.001** |
| No | 155 (52.9) | 123 (73.2) | |
| Yes | 138 (47.1) | 45 (26.7) | |
| **Abdominal bloating** | | | 0.59 |
| No | 190 (64.8) | 113 (67.2) | |
| Yes | 103 (35.1) | 55 (32.7) | |
| **Nausea** | | | 0.65 |
| No | 207 (70.6) | 122 (72.6) | |
| Yes | 86 (29.3) | 46 (27.3) | |
| **Fever** | | | 0.422 |
| No | 221 (75.4) | 121 (72.0) | |
| Yes | 72 (24.5) | 47 (27.9) | |
| **Shivering** | | | 0.28 |
| No | 229 (78.1) | 124 (73.8) | |
| Yes | 64 (21.8) | 44 (26.1) | |
| **New onset diabetes** * | | | 0.83 |
| No | 260 (88.7) | 148 (88.1) | |
| Yes | 33 (11.2) | 20 (11.9) | |
| **Steatorrhea** | | | 0.50 |
| No | 263 (89.7) | 154 (91.6) | |
| Yes | 30 (10.2) | 14 (8.3) | |

¶ P value for chi-squared test

* New onset diabetes was defined as being diagnosed with diabetes mellitus in the recent 2 years

illiterate, and did not have a partner had lower overall survival and were less likely to undergo surgical resection of the tumor compared to patients who lived in urban areas, were education,

**Table 4. Associations between different demographics and tumor characteristics with the type of received treatment among Iranian patients with pancreatic ductal adenocarcinoma.**

| Characteristics | Palliative N (%) (total n = 186) | Chemotherapy N (%) (total n = 227) | Surgery N (%) (total n = 48) | P-value ؟ |
|---|---|---|---|---|
| **Gender** | | | | 0.97 |
| Male | 113 (60.7) | 139 (61.2) | 30 (62.5) | |
| Female | 73 (39.2) | 88 (38.7) | 18 (37.5) | |
| **Formal education** | | | | < **0.001** |
| Ever | 89 (47.8) | 148 (65.2) | 37 (77.0) | |
| Never | 97 (52.1) | 79 (34.8) | 11 (22.9) | |
| **Marital status** | | | | **0.005** |
| Married | 139 (74.3) | 195 (85.9) | 43 (89.5) | |
| Single | 47 (25.2) | 32 (14.1) | 5 (10.4) | |
| **Residence** | | | | 0.60 |
| Urban | 147 (79.0) | 181 (79.7) | 41 (85.4) | |
| Rural | 39 (20.9) | 46 (20.2) | 7 (14.5) | |
| **Body Mass Index** * | | | | 0.083 |
| Normal | 8 (4.4) | 3 (1.3) | 1 (2.0) | |
| Underweight | 82 (45.5) | 95 (42.6) | 13 (27.0) | |
| Overweight | 62 (34.4) | 84 (37.6) | 20 (41.6) | |
| Obese | 28 (15.5) | 41 (18.3) | 14 (29.1) | |
| **Smoking** | | | | 0.67 |
| Never | 122 (65.5) | 143 (63.0) | 28 (58.3) | |
| Former | 19 (10.2) | 33 (14.5) | 7 (14.5) | |
| Current | 45 (24.1) | 51 (22.4) | 13 (27.0) | |
| **Opium use** | | | | 0.27 |
| Never | 149 (80.1) | 195 (85.9) | 43 (89.5) | |
| Former | 9 (4.8) | 6 (2.6) | 0 (0) | |
| Current | 28 (15.0) | 26 (11.4) | 5 (10.4) | |
| **Alcohol drinking** | | | | 0.25 |
| Never | 168 (90.3) | 205 (90.3) | 41 (85.4) | |
| Former | 11 (5.9) | 8 (3.5) | 5 (10.4) | |
| Current | 7 (3.7) | 14 (6.1) | 2 (4.1) | |
| **Tumor location** | | | | **0.016** |
| Head | 148 (79.5) | 153 (67.4) | 41 (85.4) | |
| Body | 32 (17.2) | 63 (27.7) | 7 (14.5) | |
| Tail | 6 (3.2) | 11 (4.8) | 0 | |
| **Tumor Stage** | | | | < **0.001** |
| I | 24 (12.9) | 11 (4.8) | 9 (18.7) | |
| II | 103 (55.3) | 114 (50.2) | 32 (66.6) | |
| III | 20 (10.7) | 56 (24.6) | 6 (12.5) | |
| IV | 39 (20.9) | 46 (20.2) | 1 (2.0) | |

؟ P value for chi-squared test

* Data on body mass index at enrolment is missing for 10 participants

and had a partner. Further, rural residence remained an independent prognostic factor after adjustment for other risk factors. The prognostic effects of socioeconomic status have been also shown in the studies that were conducted in the developed countries including Denmark [20], the United States [21, 22], Netherlands [6, 23], and Canada [24]. These results indicate the requirement for strategies to enhance access to equipped healthcare centers in rural areas,

**Table 5. Association between different demographical, clinical, and therapeutic factors with risk of death from pancreatic cancer.**

| Characteristic | N (%) | Adjusted Model 1 † HR (95% CI) | Model 1 † P value | Adjusted Model 2 ⊣ HR (95% CI) | Model 2 ⊣ P value |
|---|---|---|---|---|---|
| **Age (years)** | | | | | |
| Each 10 years increase in age | - | **1.11 (1.01–1.23)** | **0.023** | 1.03 (0.93–1.14) | 0.55 |
| **Gender** | | | | | |
| Female | 177 (39.2) | 1 | - | 1 | - |
| Male | 274 (60.8) | **1.31 (1.03–1.66)** | **0.024** | 1.05 (0.80–1.38) | 0.69 |
| **Formal education** | | | | | |
| Ever | 269 (59.6) | 1 | - | 1 | - |
| Never | 182 (40.3) | 1.19 (0.95–1.49) | 0.12 | 0.93 (0.72–1.20) | 0.60 |
| **Marital status** | | | | | |
| Married | 370 (82.0) | 1 | - | 1 | - |
| Single | 81 (17.9) | **1.50 (1.11–2.01)** | **0.0065** | 1.10 (0.81–1.49) | 0.50 |
| **Residence** | | | | | |
| Urban | 361 (80.0) | 1 | - | 1 | - |
| Rural | 90 (19.9) | 1.22 (0.95–1.58) | 0.11 | **1.33 (1.01–1.74)** | **0.033** |
| **Body Mass Index** | | | | | |
| Normal | 190 (42.1) | 1 | - | 1 | - |
| Underweight | 12 (2.6) | 1.11 (0.59–2.06) | 0.73 | 0.48 (0.18–1.30) | 0.17 |
| Overweight | 166 (36.8) | **0.78 (0.63–0.98)** | **0.037** | 0.88 (0.64–1.21) | 0.45 |
| Obese | 83 (18.4) | **0.70 (0.52–0.94)** | **0.020** | 0.90 (0.59–1.35) | 0.59 |
| **Symptom count** | | | | | |
| Each Additional symptom | - | **1.05 (1.01–1.08)** | **0.0069** | **1.06 (1.02–1.11)** | **0.0011** |
| **Smoking** | | | | | |
| Never | 288 (63.8) | 1 | - | 1 | - |
| Former | 58 (12.8) | 0.89 (0.64–1.22) | 0.47 | 0.97 (0.69–1.34) | 0.84 |
| Current | 105 (23.2) | 0.94 (0.72–1.22) | 0.65 | 0.81 (0.59–1.10) | 0.19 |
| **Opium use** | | | | | |
| Never | 381 (84.4) | 1 | - | 1 | - |
| Former | 15 (3.3) | 1.54 (0.87–2.71) | 0.13 | 1.50 (0.83–2.74) | 0.17 |
| Current | 55 (12.2) | **1.40 (1.03–1.91)** | **0.030** | **1.51 (1.04–2.20)** | **0.027** |
| **Drinking alcohol** | | | | | |
| Never | 404 (89.5) | 1 | - | 1 | - |
| Former | 24 (5.3) | 0.85 (0.54–1.36) | 0.52 | 0.90 (0.54–1.48) | 0.61 |
| Current | 23 (5.1) | 0.81 (0.51–1.28) | 0.38 | 0.71 (0.47–1.28) | 0.32 |
| **Tumor location** | | | | | |
| Head | 336 (74.5) | 1 | - | 1 | - |
| Body | 98 (21.7) | 1.22 (0.96–1.55) | 0.088 | **1.33 (1.02–1.75)** | **0.038** |
| Tail | 17 (3.7) | 1.14 (0.67–1.96) | 0.61 | 0.91 (0.51–1.65) | 0.77 |
| **Stage at diagnosis** | | | | | |
| I | 43 (9.5) | 1 | - | 1 | - |
| II | 244 (54.1) | 1.19 (0.84–1.69) | 0.32 | 1.32 (0.91–1.91) | 0.14 |
| III | 80 (17.7) | 1.29 (0.86–1.93) | 0.21 | 1.51 (0.97–2.34) | 0.07 |
| IV | 84 (18.6) | **2.15 (1.44–3.21)** | **0.0002** | **2.07 (1.34–3.19)** | **0.0012** |

- Only patients who did not have missing information on any of the presented variables were included in this analysis, and 10 participants who had missing information on body mass index at enrolment were removed

**HR:** Hazards Ratio, **CI:** Confidence Interval

†: This model is adjusted for age, gender, formal education, marital status, and residence.

⊣: The fully adjusted model is stratified by the received treatment and simultaneously includes all variables as shown in the table

and also the need to provide more education and support to patients who have lower socioeconomic status and are less educated, and also to those single patients who might feel less supported compared to patients who live with a partner.

In our study having more symptoms at diagnosis was an independent predictor of survival in PC patients. Consistent with our findings several retrospective studies have also suggested the prognostic effects of the number of symptoms upon diagnosis of PC [25–27]. Further, we found that abdominal pain was associated with diagnosis at higher stages, while jaundice, urine discoloration, and pruritus were associated with diagnosis at lower stages. Although abdominal is the most common symptom of pancreatic cancer, it is very nonspecific and therefore focusing only this symptom may lead to a delayed diagnosis of PC [8, 28]. However, patients who have obstructive jaundice and its associated symptoms are usually referred for thorough investigations and therefore are diagnosed earlier than patients with the nonspecific symptoms [25, 28, 29]. This was also documented in a systematic review of pancreatic cancer symptoms that found jaundice to have the highest positive predictive value for the diagnosing PC [29]. In two case-control studies, patients with PC were reported to have visited the primary healthcare centers on a median of 18 and 26 times before receiving the correct diagnosis [28, 30]. These findings show the requirement for raising awareness among primary healthcare providers to consider the diagnosis of PC in high risk older individuals with chronic non-specific symptoms.

In the current study patients who had tumors located in the head of pancreas were diagnosed at an earlier stage and had better survival than patients who had tumors located in the body and tail of pancreas. Several studies have shown that tumor location can affect the survival of PC patients [25, 31, 32], with tumors that are located in the head having the best prognosis as these patients often develop jaundice and present at early stages of the disease, while those with distal lesions often have nonspecific symptoms and remain undiagnosed until advanced stages [25, 30–32].

In this study having an advanced stage tumor was an independent risk factor for mortality. Further, while our study was an observational study that did not aim to assess the effects of different treatments on PC mortality, we found a better overall survival in patients who underwent surgery and received chemotherapy compared to those who only received palliative (symptomatic) treatment. The effects of stage at diagnosis, and surgical resection of the tumor on the survival of PC are well described in many studies [4, 8, 33]. Although in the available studies chemotherapy has been shown some beneficial effects on the short-term survival of PC patients [34], poor response to chemotherapy remains a serious problem in managing PC patients who have unresectable tumors and further efforts are needed to improve the long-term survival in these patients [3, 19].

To our knowledge our study is the first prospective study to show the independent effect of using opium on PC survival. Several retrospective studies have shown the vicious effects of opioid use on survival time in different cancer types including PC [35], gastric cancer [36], and lung cancer [37]. In two other retrospective studies of patients with advanced incurable cancers opioid use was independently associated with shorter survival [38, 39]. Many experimental studies have documented the tumor promoting effects of opioids that are important for tumor growth, invasiveness and metastasis [40]; these effects include activating angiogenesis and neovascularization [40], facilitating cancer cell proliferation and migration [40], and impairing immune functions [41]. Additionally, opioid receptors have been found in PC tissues and the experimental studies have shown that inhibiting these receptors results in the inhibition of PC progression [42, 43], as the development and progression of PC are shown to be related to opioid receptor pathways [42, 44].

The main strengths of this study are being the largest prospective survival study of pancreatic cancer in western Asia and Northern Africa, the first prospective study that assesses the

effect of using opium on PC survival, having actively followed-up PC patients from different provinces of the country on monthly basis, not having any loss to follow-up, and not having any missing clinical and pathologic information from any patient. The main limitation of this study was using EUS and conventional CT scan for evaluating the stage of PC on enrolment. Spiral CT scan and magnetic resonance imaging (MRI) have better resolutions and are more accurate in detecting small lesions, and therefore we might have underestimated the stage of PC in some patients. However, due to the prospective design of this study, any error in measuring PC stage is likely to be non-differential. Another limitation is that we did not gather the laboratory data including tumor markers and albumin levels for most patients, although the effect of these factors are controversial, having gathered these information could help in better understanding of the prognostic factors that affect the survival of PC patients in this region.

## Conclusions

Five-year survival rates of PC in Iranian patients are lower than the developed countries. Besides the stage and pathologic features of PC, socioeconomic characteristics might influence the probabilities of receiving treatment and survival in these patients. Opium use is a novel prognostic factor for PC survival in this region.

## Supporting information

**S1 File.**
(DOCX)

**S2 File.**
(DOCX)

**S1 Table. Estimates of overall survival rates by clinical symptoms upon diagnosis among pancreatic cancer patients.**
(DOCX)

**S1 Questionnaire.**
(PDF)

**S2 Questionnaire.**
(PDF)

## Acknowledgments

**Disclaimer:** Where authors are identified as personnel of the International Agency for Research on Cancer / World Health Organization, the authors alone are responsible for the views expressed in this article and they do not necessarily represent the decisions, policy or views of the International Agency for Research on Cancer / World Health Organization.

## Author Contributions

**Conceptualization:** Razieh Bakhshandeh, Alireza Moayyedkazemi, Farhad Zamani, Saman Nikeghbalian, Paul Brennan, Reza Malekzadeh, Akram Pourshams.

**Data curation:** Mahdi Sheikh, Sahar Masoudi, Razieh Bakhshandeh, Alireza Moayyedkazemi, Farhad Zamani, Sepideh Nikfam, Masoumeh Mansouri, Neda Ghamarzad Shishavan, Saman Nikeghbalian, Reza Malekzadeh, Akram Pourshams.

**Formal analysis:** Mahdi Sheikh, Sahar Masoudi, Akram Pourshams.

**Funding acquisition:** Paul Brennan, Reza Malekzadeh, Akram Pourshams.

**Investigation:** Mahdi Sheikh, Alireza Moayyedkazemi, Farhad Zamani, Sepideh Nikfam, Saman Nikeghbalian, Paul Brennan, Reza Malekzadeh, Akram Pourshams.

**Methodology:** Mahdi Sheikh, Razieh Bakhshandeh, Alireza Moayyedkazemi, Farhad Zamani, Sepideh Nikfam, Masoumeh Mansouri, Neda Ghamarzad Shishavan, Saman Nikeghbalian, Paul Brennan, Reza Malekzadeh, Akram Pourshams.

**Project administration:** Mahdi Sheikh, Sahar Masoudi, Akram Pourshams.

**Resources:** Farhad Zamani, Sepideh Nikfam, Masoumeh Mansouri, Saman Nikeghbalian, Paul Brennan, Reza Malekzadeh, Akram Pourshams.

**Software:** Mahdi Sheikh, Sahar Masoudi, Razieh Bakhshandeh, Masoumeh Mansouri, Neda Ghamarzad Shishavan.

**Supervision:** Paul Brennan, Reza Malekzadeh, Akram Pourshams.

**Validation:** Alireza Moayyedkazemi, Neda Ghamarzad Shishavan, Paul Brennan, Reza Malekzadeh, Akram Pourshams.

**Visualization:** Mahdi Sheikh, Paul Brennan, Reza Malekzadeh, Akram Pourshams.

**Writing – original draft:** Mahdi Sheikh, Sahar Masoudi, Akram Pourshams.

**Writing – review & editing:** Mahdi Sheikh, Paul Brennan, Reza Malekzadeh, Akram Pourshams.

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
