## [Decision Letter · Decision Letter 0]

7 Oct 2020

PONE-D-20-16539

Survival features, prognostic factors, and determinants of diagnosis and treatment among Iranian patients with pancreatic cancer, a prospective study

PLOS ONE

Dear Dr. Pourshams,

Thank you for submitting your manuscript to PLOS ONE. After careful consideration, we feel that it has merit but does not fully meet PLOS ONE’s publication criteria as it currently stands. Therefore, we invite you to submit a revised version of the manuscript that addresses the points raised during the review process.

We look forward to receiving your revised manuscript.

Kind regards,

Ulrich Wellner, PD Dr. med.

Academic Editor

PLOS ONE

Journal Requirements:

2. Please provide additional details regarding participant consent. In the ethics statement in the Methods and online submission information, please ensure that you have specified what type of consent you obtained (for instance, written or verbal, and if verbal, how it was documented and witnessed). If your study included minors, state whether you obtained consent from parents or guardians.

3. Please include additional information regarding the questionnaire used in the study and ensure that you have provided sufficient details that others could replicate the analyses. For instance, if you developed a questionnaire as part of this study and it is not under a copyright more restrictive than CC-BY, please include a copy, in both the original language and English, as Supporting Information.

4. To comply with PLOS ONE submission guidelines, in your Methods section, please provide additional information regarding your statistical analyses, specifically the threshold level of statistical significance used in the analysis. For more information on PLOS ONE's expectations for statistical reporting, please see https://journals.plos.org/plosone/s/submission-guidelines.#loc-statistical-reporting.

Reviewers' comments:

Reviewer's Responses to Questions

**Comments to the Author**

1. Is the manuscript technically sound, and do the data support the conclusions?

Reviewer #1: Partly

Reviewer #2: Yes

2. Has the statistical analysis been performed appropriately and rigorously? 

Reviewer #1: No

Reviewer #2: Yes

3. Have the authors made all data underlying the findings in their manuscript fully available?

Reviewer #1: Yes

Reviewer #2: Yes

4. Is the manuscript presented in an intelligible fashion and written in standard English?

Reviewer #1: Yes

Reviewer #2: Yes

5. Review Comments to the Author

Reviewer #1: Sheikh et al. investigated the survival features, and determinants of treatment and stage at presentation in Iran. The authors found socioeconomic disparities and using opium negatively impact the probabilities of receiving treatment and/or survival in these patients. The study design and the result is confused. Many points need further explanations.

Major point:

1. This study enrolled 461 pancreatic adenocarcinoma patients. Were these PC patients all pancreatic ductal adenocarcinoma? The pathological evidence is not clear. In this study, only 48 patients received surgery, however, 293 patients were with tumor stage I+II. The resection rate was relatively low. Why?

2. The basic characteristics data were not shown in all patients including gender, age, laboratory tests, etc.

3. Surgical resection was performed on 10.3% of the participants, while 56.7% received chemotherapy either with surgery, 40.3% of the participants only received palliative treatment. What is the chemotherapy regimen for these patients? Among all PC patients, 40.3% of the participants only received palliative treatment, what is the palliative treatment in this study. Why these patients received no chemotherapy or at least oral chemotherapy.

4. The author claimed that this study was the prospectively designed study, was this study a registered clinical trial? If yes, please provided the clinical trial No..

5. Some results need more explanation. Higher overall survival was observed in patients who were younger (p<0.001), educated (p=0.004), married (p=0.003), lived in urban areas (p=0.019), and had never used opium (p=0.039). Patients with younger age and married patients had significantly longer survival, why? Please explain the relationships. Moreover, all risk factors in this study all used univariate regression model, the significant risk factor couldn’t exclude the influence of confounding factors. Thus, multivariate regression model should be used, and the following significant risk factor had clinical values.

Minor points:

1. P values should be added in Table 4.

2. It is already 2020, we should use AJCC 8th edition for identifying tumor stage.

Reviewer #2: This study is interesting, the most significant point is using opium is negatively with the survival of PC patients. There are some suggestions for modification of this manuscript:

1. the introduction is too long, some contents are background information for PC, I suggested authors should delete this contents. and this introduction should be closely corrected to the paper's results and conclusions.

2. All the patients received EUS and FNA? Based on the NCCN guideline, enhanced CT scan combined with CA199 is enough for the diagnosis for PC. Why all the patient will receive this invasive testing?

3. The author indicated that "Opium use is a novel prognostic factor for PC survival in this region", also indicated "promoting effects of opioids are important for tumor growth, invasiveness and metastasis; these effects include activating angiogenesis and neovascularization, facilitating cancer cell proliferation and migration" based on published papers. As an anaesthetic drug, the PC patients using opium for long time may have a lower tumor stage because of delayed diagnosis. For example, the PC patients using opium have higher threshold of pain, which may cause delayed visiting doctor. Is this a reason for this result?

4. I suggest the authors should analysis the characteristics of the opium using group, including the tumor stage, tumor location, whether received surgery, etc.

6. PLOS authors have the option to publish the peer review history of their article (what does this mean?). If published, this will include your full peer review and any attached files.

Reviewer #1: No

Reviewer #2: No

---

## [Author Response · Author response to Decision Letter 0]

9 Nov 2020

Dear Editor and Reviewers

We would like to thank you very much for your precious time and helpful comments that helped to enhance the quality of the paper. Reading some of your comments, we recognized that there is some confusion and misunderstanding on the study aims and design. Therefore, we made major revisions to the used models, tables, results, and discussion to prevent this confusion and focus on the study aims and original questions. Briefly, we made the following revisions to the manuscript: 

1. In the main adjusted models (Table5), instead of assessing the effects of treatments in the adjusted models (which was not among the aims of the study), we stratified the models by the received treatments. This way we account for the effects of treatments (which had time varying effects) without investigating their efficacy. Further, we also assessed the effects of BMI at enrolment and adjusted the results for BMI at enrolment. We would like to emphasize that despite all the mentioned modifications to the used models, the main results remained very similar to the previous version. However, the presentation of the results have significantly improved.

2. We combined tables 1 and 3 in the previous version in one table (Table 1 in the current revision). Also in this table, we removed the 95%CI column for the median survival to only show the observed values (rather than making any estimations).

3. We moved Table 2 in the previous version to the supplementary material and renamed this table as Table S1. This table shows the prevalence of symptoms on enrolment and also illustrates the overall survival rates based on the symptoms. We have already presented the results of this table in the text under Results - Clinical symptoms section of the manuscript.

4. We moved the supplementary tables in the previous versions to the main manuscript. These tables show the association between different demographics, exposures, and tumor factors with tumor stage at diagnosis (Table 2 in the current revision), the association between clinical symptoms with tumor stage at diagnosis (Table 3 in the current revision), and the association between different demographics, exposures, and tumor factors with the received treatment at the follow-up (Table 4 in the current revision). These tables show the determinants of the tumor stage at diagnosis and the received treatment in the study population and were among the study aims. Therefore, they were moved to the main text.

5. We removed Figure 1 in the previous version that only visualized some factors of Table 1 due to showing repetitive results. However, we added a new figure (Figure 1 in the current revision) that shows the overall survival curves during the first 12 months after diagnosis that are stratified by education, residence, and opium use which are among the most important findings of the paper and have not been presented in other tables.

6. Based on the journal’s guidelines, we placed each table in the manuscript file directly after the paragraph in which it is first cited, rather than including them altogether at the end of the manuscript.

In addition to the above-mentioned revisions we made some more minor revision based on the suggestions from the reviewers.

Please find below point by point responses to the comments and the revisions that we made to the paper accordingly.

Reviewer #1

Sheikh et al. investigated the survival features, and determinants of treatment and stage at presentation in Iran. The authors found socioeconomic disparities and using opium negatively impact the probabilities of receiving treatment and/or survival in these patients. The study design and the result is confused. Many points need further explanations.

Major point:

1. This study enrolled 461 pancreatic adenocarcinoma patients. Were these PC patients all pancreatic ductal adenocarcinoma? The pathological evidence is not clear. In this study, only 48 patients received surgery, however, 293 patients were with tumor stage I+II. The resection rate was relatively low. Why?

Response

• Yes, patients were enrolled if they had a confirmed diagnosis of pancreatic ductal adenocarcinoma. We modified the manuscript to clarify this.

• For further clarifications on the pathological evidence we added the following explanations to lines 89-94 on page 4 of the manuscript as follows “The participants were then referred for collection of bio-samples and performing EUS, and in case of finding a mass or cystic lesions, they underwent fine needle aspiration (FNA). The obtained samples were then reviewed by one expert pathologist who was blinded to the questionnaire data. If the diagnosis of ductal adenocarcinoma could not be finalized by hematoxylin/eosin staining, the samples were evaluated using an immunohistochemistry panel to differentiate ductal adenocarcinoma from other tumors.”

• We would like to emphasize that this study is a prospective observational study to investigate the survival and prognostic features of pancreatic cancer, and also the determinants of receiving treatment for this cancer in a middle income country. It is not an interventional study and we just followed the patients to collect information on their vital status and their treatments and procedures. As we have emphasized in the last paragraph of the introduction, one of the main aims for this study is “to address the unmet medical needs and the possible shortfalls in the management of PC in this region”. As the reviewer mentioned this study showed the surgical resection rate is relatively low in this country (which might be the case with other low and middle income countries). This is an important finding of this study and we have already emphasized this finding in the Discussion section, lines 331-332, page 17, of the manuscript as follows “These results show the need for more efforts to improve the clinical management of PC patients in this region and to design more studies aiming to investigate the underlying reasons of the observed poor long-term survival in PC patients with early stage tumors”.

• Regarding the reasons behind the low surgery rates, this might be due to the fact that in low and middle income countries there are no fast track management procedures for pancreatic cancer patients and also many patients do not have access to equipped health care centers. Therefore, some patients might be diagnosed with an early stage pancreatic cancer tumor but because of the unavailability of specialized surgeons and surgery unites their disease might progress rapidly which in turn could affect the clinical management of this patients. We have already performed an analysis to investigate the determinants of receiving treatments (including surgery) in this population that is presented in Table 4 of the current revision (Supplementary Table2 in the previous version) and also in the second paragraph of the Results - Determinants of stage and the probability of receiving treatment section of the paper. We have also emphasized these concerning results in the 3rd paragraph of Discussion section, pages 16 and 17 of the manuscript as “In the current study several indicators of socioeconomic status were related to survival and the probability of receiving treatment in PC patients. Patients who lived in rural areas, were illiterate, and did not have a partner had lower overall survival and were less likely to undergo surgical resection of the tumor compared to patients who lived in urban areas, were education, and had a partner … These results indicate the requirement for strategies to enhance access to equipped healthcare centers in rural areas, and also the need to provide more education and support to patients who have lower socioeconomic status and are less educated …”

• Another probable reason might be the underestimation of disease stage in some patients in this study, which we have already mentioned as a limitation of this study in the last paragraph of the Discussion section, lines 374-378, page 18 as follows “The main limitation of this study was using EUS and conventional CT scan for evaluating the stage of PC on enrolment. Spiral CT scan and magnetic resonance imaging (MRI) have better resolutions and are more accurate in detecting small lesions, and therefore we might have underestimated the stage of PC in some patients. However, due to the prospective design of this study, any error in measuring PC stage is likely to be non-differential.”

2. The basic characteristics data were not shown in all patients including gender, age, laboratory tests, etc.

Response:

• Please kindly note that the basic characteristics data (age, gender, education, marital status, residence), tumor characteristics, and also some exposure data (smoking, opium use, alcohol intake) for all patients have been already shown in the first 2 columns of Tabl1. Further, the clinical symptoms have also been shown for all patients in the first 2 columns Tables S1 in this revision (Table 2 in the previous version). Finally, in the Results – Demographics section of the paper the demographics are again shown as mean ± standard deviations and percentages.

• Assessing the effects of laboratory tests and tumor markers were not among the aims of this study and we have already mentioned this issue in the last paragraph of the Discussion section as follows “Another limitation is that we did not gather the laboratory data including tumor markers and albumin levels for most patients, although the effect of these factors are controversial, having gathered these information could help in better understanding of the prognostic factors that affect the survival of PC patients in this region.”

3. Surgical resection was performed on 10.3% of the participants, while 56.7% received chemotherapy either with surgery, 40.3% of the participants only received palliative treatment. What is the chemotherapy regimen for these patients? Among all PC patients, 40.3% of the participants only received palliative treatment, what is the palliative treatment in this study. Why these patients received no chemotherapy or at least oral chemotherapy.

Response:

• As we mentioned earlier, this is an observational study that aimed to investigate the survival and prognostic features of pancreatic cancer in this region. The study did not aim to assess the effects of different regimens and treatments, rather it aimed to investigate the determinants of receiving any type of treatment. The aims of this study have been clearly mentioned in the last paragraph of Introduction as “In this prospective study we analyze the clinical, pathological, therapeutic, and survival features of 461 histologically confirmed PC cases to provide reliable information on different features of PC in Iranian patients and to address the unmet medical needs and the possible shortfalls in the management of PC in this region.”

• Based on the study aims and objectives, this study was not designed to investigate the effect of different chemotherapy regimens which would needed designing an interventional study (a clinical trial) with different sample size numbers and methods. Accordingly, based on the aims of this observational study, the patients were considered as received chemotherapy if they had received any chemotherapy regimen. We wanted to investigate what are the demographics of patients who are more likely to receive (or have access to) chemotherapy (regardless of the regimen) or surgery in this region.

• The palliative treatments in this study were the treatments that were given to patients to alleviate the symptoms including the insertion of biliary stent for the jaundice, pain treatments, etc.

• This study is a real life observational study to address pancreatic cancer survival features and management in a middle income country. The investigators did not intervene in the management and treatments of patients. The selection to receive any kind of chemotherapy or not was made by the primary physicians through consultations with their patients. Based on the physician’s explanations, disease stage, treatment efficacy, comorbidities, economic issues, etc. the patients could accept or refuse the chemotherapy. Given the nature of this study, only the information on the received treatments was gathered and no intervention was made by the study investigators.

4. The author claimed that this study was the prospectively designed study, was this study a registered clinical trial? If yes, please provided the clinical trial No.

Response:

• Please kindly note that this is an Observational Prospective study and not an Interventional or Clinical Trial study to need a registration. As mentioned earlier we did not intervene in the management or treatments of the patients. We only followed the patients to record any procedure/intervention that was done for them, and also to record the progression of the disease and the vital status of the patients.

• The purpose of collecting data on treatments during the follow-up was to 1) adjust or modify the models based on the received treatments and 2) estimate the determinants of receiving treatments. Assessing the effects of each treatment was not an aim of this study.

• Reading the comments from the reviewers we recognized that the presentation of the study results, models, and analyses created some confusion. Therefore, rather than including and presenting the treatments as separate variables in the adjusted models (which might be incorrectly interpreted as assessing the effects of treatments) in this revision we stratified the models by the received treatments to better show the study aims and design. The related Results, Tables, and Discussion were modified accordingly. Please find the full summary of the major revisions made, at the beginning of the Response to Reviewers document.

5. Some results need more explanation. Higher overall survival was observed in patients who were younger (p<0.001), educated (p=0.004), married (p=0.003), lived in urban areas (p=0.019), and had never used opium (p=0.039). Patients with younger age and married patients had significantly longer survival, why? Please explain the relationships. Moreover, all risk factors in this study all used univariate regression model, the significant risk factor couldn’t exclude the influence of confounding factors. Thus, multivariate regression model should be used, and the following significant risk factor had clinical values.

Response:

• Please kindly note that in this study two multivariate regression models were used that have already been presented as Adjusted Model 1 and Adjusted Model 2 in Table5 of this revision (Table 4 in the previous version). Also in the Footnote of Table5 we describe each model and its adjustments.

• In the Methods – Statistical Analysis section of the study, on lines 151-160, page 6, the use of multivariate model and its adjusted factors are described in details as follows “The effects of demographical, clinical, and tumor characteristics on the survival time were tested using two multivariate Cox regression models. The first multivariate model (model-1) was adjusted for the demographics including age (continuous), sex (male/female), formal education (ever/never), marital status (married/single), and residence (urban/rural), while the second multivariate model (model-2) further included BMI at enrolment (underweight/normal/overweight/obese), number of symptoms upon diagnosis (continuous), smoking cigarettes (never/former/current), using opium (never/former/current), drinking alcohol (never/former/current), tumor location (head/body/tail), and stage of PC (I/II/III/IV). The final models were further stratified by the received treatments (palliative / chemotherapy / surgery) that violated the proportional hazard (PH) assumptions by showing time-varying effects. The PH assumption was tested using Schoenfeld’s global test.”

• We tried to focus in the Discussion on the factors that showed significant prognostic values in the fully adjusted models. Age and marital status did not show significant effects in the multivariate adjusted models. However, in the third paragraph of the Discussion Section, pages 16 and 17, we have included some possible explanations for the observed effect of marital status as follows “In the current study several indicators of socioeconomic status were related to survival and the probability of receiving treatment in PC patients. Patients who lived in rural areas, were illiterate, and did not have a partner had lower overall survival and were less likely to undergo surgical resection of the tumor compared to patients who lived in urban areas, were education, and had a partner. Further, rural residence remained an independent prognostic factor after adjustment for other risk factors. The prognostic effects of socioeconomic status have been also shown in the studies that were conducted in the developed countries including Denmark [20], the United States [21,22], Netherlands [6,23], and Canada [24]. These results indicate the requirement for strategies to enhance access to equipped healthcare centers in rural areas, and also the need to provide more education and support to patients who have lower socioeconomic status and are less educated, and also to those single patients who might feel less supported compared to patients who live with a partner.” 

Minor points:

1. P values should be added in Table 4.

Response:

As suggested by the reviewer, we added the p values to Table5 in this revision.

2. It is already 2020, we should use AJCC 8th edition for identifying tumor stage.

Response:

Please kindly note that as it is mentioned in the Materials and Methods - Study population and design section, “participants were recruited … between January 2011 and January 2018”. At the time of study design and initiation of the recruitment, AJCC 8th was still not introduced. Therefore, the data were recorded based on the previous AJCC version that was the latest version of AJCC on the time of study initiation and recruitment. 

Reviewer #2

This study is interesting, the most significant point is using opium is negatively with the survival of PC patients. There are some suggestions for modification of this manuscript:

1. the introduction is too long, some contents are background information for PC, I suggested authors should delete this contents. and this introduction should be closely corrected to the paper's results and conclusions.

Response:

As suggested by the reviewer, we shortened the introduction by removing basic and irrelevant information and their corresponding references. We also modified the introduction to focus on the manuscript results and conclusions and accordingly added two very recent relevant references (References 12 and 13).

2. All the patients received EUS and FNA? Based on the NCCN guideline, enhanced CT scan combined with CA199 is enough for the diagnosis for PC. Why all the patient will receive this invasive testing?

Response:

In response to this comments we would like to mention two points:

• As the reviewer mentioned, NCCN guideline indicates that enhanced CT scan combined with CA199 can be used for the diagnosis of PC. However, the guidelines in each region is dependent on the modalities and settings in that specific region. Like many low and middle income countries, in our region enhanced CT scan is not widely available (as we have already mentioned in the limitation of the study) and the patients usually undergo conventional CT scan which does not have the same accuracy and resolution. Therefore, in LMICs the NCCN guidelines cannot be used. Further, in the current guidelines in our region to patients should have a confirmed pathologic/histologic diagnosis of PC to be considered eligible for receiving treatments.

• As we have mentioned in the Materials and Methods - Study population and design section, this study included patients who were suspicious for having a pancreatic mass and were referred by their primary physician for performing endoscopic ultrasonography (EUS) with biopsy from pancreatic tissue. Therefore, the investigators did not intervene in any process and rather designed an observational study to address the unmet medical needs and the possible shortfalls in the management of PC in this region

3. The author indicated that "Opium use is a novel prognostic factor for PC survival in this region", also indicated "promoting effects of opioids are important for tumor growth, invasiveness and metastasis; these effects include activating angiogenesis and neovascularization, facilitating cancer cell proliferation and migration" based on published papers. As an anaesthetic drug, the PC patients using opium for long time may have a lower tumor stage because of delayed diagnosis. For example, the PC patients using opium have higher threshold of pain, which may cause delayed visiting doctor. Is this a reason for this result?

Response:

We thank the reviewer for raising this important issue. To address the possible confounding effects related to opium use we used several methods: 1) assess the effects of opium use in the multivariate cox regression models that were adjusted for all potential confounders and risk factors including tumor stage, the received treatment, socioeconomic status etc. 2) performed a sensitivity analysis by removing patients who started using opium in the recent 2 years before diagnosis (to address the possibility of reverse causality). 3) assessed tumor stage at diagnosis between opium users and non-users (Table 2 in the current revision, Table S2 in the previous version). 4) assessed the possibility of receiving treatment between opium users and non-users (Table 4 in the current revision, Table S1 in the previous version). In all the analyses the effect of opium remained constant and we did not detect any difference in the tumor stage and treatments between opium users and non-users. Therefore, while like any observational study we cannot rule out the possible effects from residual confounding, we believe that the observed effects might possibly be due to the harmful effects of using opium. 

4. I suggest the authors should analysis the characteristics of the opium using group, including the tumor stage, tumor location, whether received surgery, etc.

Response:

As suggested by the reviewer, in this revision we moved Tables S1 and S2 from the supplementary files to the main text and renamed them as Tables 2-4 in the current revision. These tables show the tumor stage and the received treatments between different strata of the demographics and some exposures including opium use. In addition, we confirm that tumor location was not different between opium users and nonusers.

---

## [Editor Report · Decision Letter 1]

23 Nov 2020

Survival features, prognostic factors, and determinants of diagnosis and treatment among Iranian patients with pancreatic cancer, a prospective study

PONE-D-20-16539R1

Dear Dr. Pourshams,

We’re pleased to inform you that your manuscript has been judged scientifically suitable for publication and will be formally accepted for publication once it meets all outstanding technical requirements.

Kind regards,

Ulrich Wellner, PD Dr. med.

Academic Editor

PLOS ONE
---

## [Editor Report · Acceptance letter]

26 Nov 2020

PONE-D-20-16539R1 

Survival features, prognostic factors, and determinants of diagnosis and treatment among Iranian patients with pancreatic cancer, a prospective study 

Dear Dr. Pourshams:

I'm pleased to inform you that your manuscript has been deemed suitable for publication in PLOS ONE. Congratulations! Your manuscript is now with our production department. 

Kind regards, 

on behalf of

Dr. Ulrich Wellner 

Academic Editor

PLOS ONE